# FORB: A Flat Object Retrieval Benchmark for Universal Image Embedding

**Pengxiang Wu, Siman Wang, Kevin Dela Rosa, Derek Hao Hu**
Snap Inc.
{pwu,swang7,kevin.delarosa,hao.hu}@snap.com

## Abstract

Image retrieval is a fundamental task in computer vision. Despite recent advances in this field, many techniques have been evaluated on a limited number of domains, with a small number of instance categories. Notably, most existing works only consider domains like 3D landmarks, making it difficult to generalize the conclusions made by these works to other domains, *e.g.*, logo and other 2D flat objects. To bridge this gap, we introduce a new dataset for benchmarking visual search methods on flat images with diverse patterns. Our flat object retrieval benchmark (FORB) supplements the commonly adopted 3D object domain, and more importantly, it serves as a testbed for assessing the image embedding quality on out-of-distribution domains. In this benchmark we investigate the retrieval accuracy of representative methods in terms of candidate ranks, as well as matching score margin, a viewpoint which is largely ignored by many works. Our experiments not only highlight the challenges and rich heterogeneity of FORB, but also reveal the hidden properties of different retrieval strategies. The proposed benchmark is a growing project and we expect to expand in both quantity and variety of objects. The dataset and supporting codes are available at https://github.com/pxiangwu/FORB/.

## 1 Introduction

Image retrieval is a fundamental and long-standing task in computer vision. Given a query image, this task aims to search for the most similar images from a large database. Recent methods have achieved remarkable performance on certain domains, such as 3D landmark [37, 49] and clothes [25]. To perform image retrieval, the prevailing practice is to map the query image into a compact embedding space, where similar images are close to each other while dissimilar ones are separated away. This embedding space can be handcrafted and one classic design is the Bag of Words (BoW) [2, 8]. A more effective idea is to learn the embedding automatically, based on deep neural netowrks [11, 3, 29, 38]. However, all these methods have only been evaluated on a limited number of domains (*e.g.*, 3D landmarks), and as a result, it remains unclear if the embedding of one method is more general than the others. In particular, for learning-based methods, since they are usually trained on a specific restricted dataset with limited object classes (*e.g.*, ImageNet [9] and Open Images [21]), their feature embeddings could be not universal enough to generalize to various open-world objects. Therefore, it is necessary to have benchmarks supplementary to the existing ones for a more comprehensive evaluation of the embeddings, especially in terms of their out-of-distribution (OOD) generalization ability.

In particular, existing image retrieval benchmarks mainly involve domains of 3D objects. Examples of the commonly considered objects include 3D landmarks, clothes, natural living things and online products. While many recent benchmarks that curate the images of these objects have sufficiently large query image sets, they are typically limited to a small number of object categories or instances. Moving beyond 3D objects, there are several datasets focusing on 2D flat objects. However, these

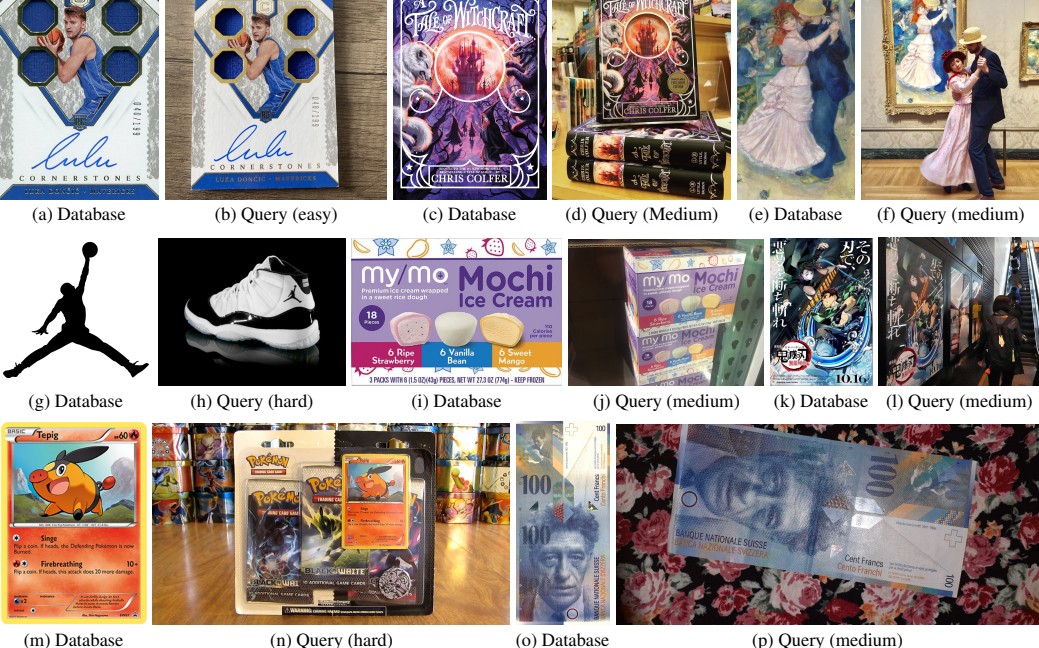

Figure 1: Example database and query images from our FORB benchmark. For each query image, we show its corresponding index image and the retrieval difficulty. The images are from different content domains: (a)(b) photorealistic trading card; (c)(d) book cover; (e)(f) painting; (g)(h) logo; (i)(j) packaged goods; (k)(l) movie poster; (m)(n) animated trading card; (o)(p) currency.

datasets are mostly small in size and related to one particular type of object, *i.e.*, logo [41, 45]. Besides, their query images tend to be in canonical pose without much distraction from the background, making the retrieval less challenging.

In order to fill the domain gap of existing benchmarks and to encourage future research in this area, we present a Flat Object Retrieval Benchmark (FORB) which contains diverse flat objects with different query difficulties. The flat objects are those with 2D surface only, which bears the textures and patterns of the object (*e.g.*, painting and logo; see Figure 1). Despite being one dimension less than 3D objects, such flat surfaces still pose many challenges for image retrieval. In particular, there can be large variations between the query and database images, due to surface and color distortions, perspective transformation, view occlusion, and illumination change. Our benchmark takes into account all these challenges and covers objects with a variety of textures (see Section 3). Notably, these objects are common in daily life and our benchmark could benefit diverse real-world visual search applications, such as recognizing logos for brand promotion, augmenting artwork exhibits in a museum, online shopping and more.

To understand how different image embeddings perform on our benchmark, we evaluate the retrieval accuracy from two perspectives: (1) Candidate rank, which corresponds to the sorted order of database images based on their similarities to the query image. The correctness of ranks reflects the discriminative ability of image embedding and can be measured with mean Average Precision (mAP). (2) Matching score margin. For a query image, ideally its matching scores against ground-truth database images should be high (*e.g.*, assuming cosine similarity), while the scores against non-relevant images should be low. Therefore, the degree of compliance with this ideal margin also delineates the quality of image embedding, a viewpoint which is largely ignored by previous works. To measure this margin, we propose to query the given image against distractor images, giving *false positive* candidates. By thresholding the matching scores, we can compute a specific false positive rate (FPR) and an updated mAP, which together quantify the margin of image embeddings. In particular, an ideal embedding should have a low FPR while keeping a high mAP.

To establish baselines on our benchmark, we evaluate a series of representative methods, including both learning-based models and handcrafted designs. Our results reveal intriguing properties of

Table 1: Comparison of our benchmark against existing image retrieval datasets.

| Dataset | Domain | # Query | # Database | Has distractor | Has difficulty label |
|---|---|---|---|---|---|
| Oxford [33] | 3D landmark | 55 | 5K | ✗ | ✗ |
| Paris [35] | 3D landmark | 55 | 6K | ✗ | ✗ |
| $\mathcal{R}$-Oxford [37] | 3D landmark | 70 | 5K + 1M | ✓ | ✓ |
| $\mathcal{R}$-Paris [37] | 3D landmark | 70 | 5K + 1M | ✓ | ✓ |
| GLD [30] | 3D landmark | 118K | 1.1M | ✓ | ✗ |
| GLDv2 [49] | 3D landmark | 118K | 762K | ✓ | ✗ |
| CUB [48] | Bird | 6K | 6K | ✗ | ✗ |
| Cars196 [20] | Car | 8K | 8K | ✗ | ✗ |
| SOP [31] | 3D product | 60K | 60K | ✗ | ✗ |
| DeepFashion [25] | Clothes | 14K | 13K | ✗ | ✗ |
| VehicleID [24] | Vehicle | 35.6K | 4.8K | ✗ | ✗ |
| iNaturalist [46] | Plant & Animal | 136K | 136K | ✗ | ✗ |
| FlickrLogos [41] | Flat object (logo) | 4K | 320 | ✓ | ✗ |
| FORB | Flat object | 14K | 54K | ✓ | ✓ |

embeddings built from different feature levels. Specifically, we show that even a model is trained on 3D objects, its embedding induced from low- or mid-level image features can still be universal enough to distinguish diverse flat objects. Moreover, for feature-scarce images, embeddings based on high-level features tend to achieve better accuracy.

Our contributions include: (1) We introduce FORB, a new visual search benchmark for evaluating image embeddings on flat objects. FORB supplements the commonly used 3D object benchmarks and essentially provides a platform for assessing the OOD generalization ability of an embedding method. (2) We propose a new evaluation metric motivated by matching score margin. This metric is complementary to mAP and offers a new perspective on image embedding quality. (3) We conduct comprehensive comparisons for different representative methods, providing solid baselines for future method developments. (4) Our evaluation results reveal the hidden properties of different retrieval strategies as well as their limitations, providing insights into the development of new techniques.

## 2  Related Work

### 2.1  Existing Datasets and Benchmarks for Image Retrieval

There has been a long history of developing benchmarks for image retrieval. For example, to promote research in instance-level recognition and search, Oxford [33] and Paris [35] datasets were introduced and have motivated a wealth of innovations in this field. With a similar motivation, researchers curated CUB [48] and Cars196 [20] to facilitate fine-grained object matching. Despite the popularity of these datasets, they are small in size and only involve a limited number of instances and categories. To further enrich the object domains for image retrieval and increase the size and complexity of the task, several more challenging datasets were constructed, such as SOP [31], DeepFashion [25], VehicleID [24], iNaturalist [46], and Google Landmarks dataset v2 (GLDv2) [49]. In particular, GLDv2 has gained widespread attention since being introduced due to its significant scale and variability, and serves as a solid benchmark for testing emerging retrieval techniques.

One limitation of these datasets is that they only focus on the task of 3D object retrieval, involving a restricted number of object domains (*e.g.*, 3D landmarks). In fact, compared to 3D objects, there exist few benchmarks on other domains, especially 2D flat objects. In real-world visual search applications, flat objects also make up a large fraction of queries. However, there are only few benchmarks on such objects and most of them are for logo [41, 45]. To fill this domain gap, our FORB benchmark includes a variety of flat objects and supplements existing 3D object benchmarks. In particular, FORB effectively serves as an OOD query set for evaluating the embeddings trained on 3D objects. In Table 1 we compare FORB against existing image retrieval datasets in detail.

It is worth mentioning that there exists another similar benchmark for assessing the generalization abilities of image embeddings, *i.e.*, Google Universal Image Embedding Challenge[1]. However, this benchmark mainly involves 3D objects and its evaluation data is kept private. We believe our FORB

---

[1]https://www.kaggle.com/competitions/google-universal-image-embedding

supplements this benchmark and will facilitate the development of visual search applications, such as organizing photo collections, visual commerce and more.

## 2.2 Out-of-Distribution Query

Most existing benchmarks only have "on-topic" queries without considering the out-of-distribution ones. As a result, they fail to present real-world challenges and are not enough to fully evaluate the quality of an image embedding. Notably, in a generic visual search app, the system tends to be queried with a large number of irrelevant queries, *i.e.*, OOD queries, for which it is expected to not yield any results. Therefore, OOD queries provide an additional important view into the robustness of image embeddings. This issue of lacking OOD queries in existing benchmarks was recognized in GLDv2 [49] and addressed with plenty of non-landmark queries. In practice, to assess the discriminative ability of image embeddings between true positive and false positive candidates, GLDv2 employs micro Average Precision ($\mu$AP), which both measures ranking performance and penalizes false positive predictions. Our FORB benchmark shares a similar motivation to GLDv2, but with a few key differences: (1) We do not provide additional OOD queries with respect to the database images. Instead, we split database into index images and distractors, and query the images against distractors. In this way we effectively turn all the query images into OOD queries. (2) Instead of using $\mu$AP, we propose a new metric, $t$-mAP, which computes an averaged mAP over different confidence thresholds. The thresholds are determined through quantiles of false positive rates. Compared to $\mu$AP and mAP, our $t$-mAP takes into account the matching score margin, which directly reflects the discriminability of image embeddings.

## 2.3 Universal Image Embedding

The quality of image embeddings determines the performance of modern image retrieval methods. Based on the design of image features, existing embeddings can be divided into two categories: handcrafted and learning-based. The former one builds image embeddings based on handcrafted low-level features (*e.g.*, SIFT [26]), using a bag of words (BoW). This design paradigm dominates many classic methods, such as [34, 27, 2, 18, 44], and usually leads to embeddings that generalize well over various domains. With the rapid advancement of deep learning, such handcrafted embeddings have been replaced with the learning-based ones in the community. The learning of image embeddings is commonly conducted in a supervised manner, on crowd-labeled datasets [15, 17, 16, 10]. However, supervised learning is not scalable since manual annotation of large-scale training data is time-consuming and costly. As a result, the training data usually contains limited pre-defined object classes (*e.g.*, ImageNet [9] and Open Images [21]), and embeddings learned from these data are not universal enough to generalize to various open-world objects [1]. In recent years, self- and weakly-supervised learning have gained extensive attention due to their less reliance on labeled data. By designing appropriate pre-text tasks and training strategies (*e.g.*, image-text matching), these learning paradigms can easily leverage a large number of unlabeled or noisy data, producing image embeddings of greater generality than supervised learning [14, 13, 7, 6, 12, 40, 19].

## 3 The FORB Benchmark

Our FORB benchmark only provides testing query images without training data. It serves as a testbed supplementary to existing benchmarks, with the following goals.

**Goals**  Our proposed benchmark aims to enrich the object domains considered in image retrieval tasks and measure the generalization ability of embedding models with respect to out-of-distribution queries. Besides, we also seek to understand the effects of image features from different levels on the embedding quality, thereby shedding light on future development of embedding models.

### 3.1 Data Collection

There are 8 different types of flat objects involved in our benchmark: (1) *Animated trading card.* We consider one particular type of card, *i.e.*, Pokemon trading card. (2) *Photorealistic trading card.* We consider cards for different sports, such as baseball, basketball, and football. (3) *Book cover*, which comes from books in different languages, such as English and Chinese. (4) *Painting*, which involves

Table 2: Overview of the proposed FORB benchmark.

| Object Type | # Query | # Index | # Distractor | # Easy | # Medium | # Hard |
|---|---|---|---|---|---|---|
| Animated trading card | 6,025 | 1,392 | 11,137 | 714 | 4,868 | 443 |
| Photorealistic trading card | 2,187 | 484 | 521 | 67 | 2,039 | 81 |
| Book cover | 1,461 | 470 | 10,739 | 66 | 1,277 | 118 |
| Painting | 988 | 430 | 615 | 119 | 710 | 159 |
| Currency | 758 | 395 | 1,188 | 112 | 576 | 70 |
| Logo | 1170 | 535 | 174 | 24 | 957 | 189 |
| Packaged goods | 800 | 476 | 2,382 | 24 | 727 | 49 |
| Movie poster | 512 | 403 | 23,094 | 49 | 426 | 37 |
| Total | 13,901 | 4,585 | 49,850 | 1,175 | 11,580 | 1,146 |

various styles, such as impressionism and baroque, etc. (5) *Currency*, which involves banknotes of modern and antique designs from different countries. We consider both the front and back of a banknote. (6) *Logo*. We consider common logos (*e.g.*, Nike) as well as long-tailed logos (*e.g.*, brands of local small businesses). (7) *Packaged goods*. We only consider products for which the corresponding index images are displayed on flat surface. (8) *Movie poster*. We consider posters from different countries, such as America and Japan. In Figure 1 we show examples for each object. As can be seen, these objects have diverse textures, involving animation and artificial patterns, etc, and thus offer various retrieval challenges. Also, they are common in daily life and retrieving such objects serves as a practical use case in real applications. For example, eBay builds an image retrieval system[2] for trading cards to facilitate the sales of cards.

To build our benchmark, we collected the query and index images mainly via Google Images. Specifically, before collecting images, for each type of objects we firstly curated a list of object names. Their names can be obtained from dedicated websites, such as TCGplayer[3] for animated trading card and Wikimedia Commons[4] for painting. Next, we queried Google Images with each of the names and retrieved the corresponding query and index images. The returned results were typically noisy and we manually filtered out the irrelevant images as well as those that could be copyright protected. In this way, we effectively matched each index image with diverse query images, giving image-level ground truths. Note that our collected query images are in the wild whereas the database images are in canonical pose (see Figure 1). Besides Google Images, we also leveraged some other sources to further augment the benchmark, such as Google Lens API, eBay, and Amazon.

To increase retrieval difficulty and challenge, similar to previous works [37, 49] we also introduced distractors to the benchmark. The distractors are images that share similar semantics, contents, or textures with the index images. They can be from the same domains as the index images, or from other domains. Distractors are primarily introduced to increase the retrieval difficulty, as they would bring perplexing features that deceive retrieval algorithms and reduce the accuracy of retrieval results. Ideally, a strong retrieval algorithm should be robust against distractors. In our benchmark, the distractor images were all from the 8 object domains and crawled from different specific websites, such as TCGplayer and Wikimedia Commons. See supplementary material for some examples. The details of our benchmark can be found in Table 2.

### 3.2 Data Annotation and Metadata

As mentioned above, we provide image-level retrieval ground truths for each query image. To enable a more detailed evaluation on the quality of image embeddings, we also offer annotations on the retrieval difficulties for each query image. Specifically, we break down difficulty into three levels: easy, medium, and hard. The specific difficulty level for a query image is subject to the following factors: (1) occlusion; (2) blur; (3) truncation; (4) color distortion; (5) perspective distortion; (6) texture complexity; (7) area of the object in the query image. For example, if the target object only occupies a small area in the image, we tag "hard" for the given query image due to the distraction of background; see Figure 1(h)(n). Similarly, if the object does not bear severe perspective distortion or truncation, we tag "easy"; see Figure 1(b). In practice, assigning difficulty levels to query images can

---

[2]https://pages.ebay.com/scantolist/

[3]https://www.tcgplayer.com/

[4]https://commons.wikimedia.org/

be a subjective process. To reduce bias and ensure precise difficulty assessment, we involve different annotators in manually labeling the difficulty of each image and then use majority voting to determine the final difficulty level. As shown in Table 4, the annotated difficulty levels are quite consistent with retrieval accuracies for all methods, *i.e.*, the accuracies are high on easy queries, whereas they are low on hard queries.

We store the annotations with a newline delimited JSON file, where each line contains the metadata corresponding to a query image. Specifically, each line is comprised of the following information: (1) query image ID; (2) the file name of query image; (3) the source URL of query image; (4) the file names of ground-truth index images; (5) the source URLs of ground-truth index images; (6) difficulty level. Here the source URL corresponds to where we downloaded the image. Apart from this annotation information, we also provide newline delimited JSON files for tracking the set of query and database images, respectively, where each line contains information regarding the image file name and source URL.

We host the metadata files at https://github.com/pxiangwu/FORB/, which is publicly accessible. As for the query and database images, they can be downloaded via the provided source image URLs. Alternatively, these images are also accessible from a Google drive, where we snapshot all the images from source URLs. Both the metadata and image files are licensed under CC BY-NC-SA.

### 3.3 Metrics

Our FORB benchmark uses the commonly adopted mAP metric, as well as a new one that takes into account the matching score margin.

**mAP** The mean Average Precision metric considers both the true positives and false positives in the ranked retrieval results. The metric is defined as follows:

$$\text{mAP@}k = \frac{1}{Q} \sum_{q=1}^{Q} \text{AP@}k(q), \quad \text{AP@}k(q) = \frac{1}{\min(m_q, k)} \sum_{k=1}^{\min(n_q, k)} \text{P}_q(k) \text{rel}_q(k), \tag{1}$$

where $Q$ is the total number of query images; $m_q$ is the number of ground-truth index images matched with query image $q$; $n_q$ is the number of predictions made by the retrieval method; $\text{P}_q(k)$ is the precision at rank $k$ for query image $q$; and $\text{rel}_q(k)$ is a relevance indicator function which equals 1 if the result at rank $k$ is relevant and equals to 0 otherwise. Note that for some query images (*e.g.*, OOD images) they do not have associated index images to retrieve, and mAP does not penalize the method even if it retrieves some results for the query images.

*t*-**mAP** To take into account OOD queries and false positive results, we introduce thresholded mAP, *i.e.*, *t*-mAP. This metric measures the matching score margin with the aid of OOD queries, and is computed as below:

$$t\text{-mAP} = \frac{1}{\tau(1)} \int_0^{\tau(1)} \text{mAP}(t) dt, \tag{2}$$

where $\tau(x)$ is the threshold that leads to a false positive rate of $1-x$ on OOD queries after thresholding the retrieved candidates with respect to their matching scores; formally, $\tau(x) = \min\{\tilde{x} \mid \text{FPR}(\tilde{x}) = 1 - x\}$, where $\text{FPR}(\tilde{x})$ is the false positive rate at threshold $\tilde{x}$. $\text{mAP}(t)$ is the mAP computed after the retrieval results are suppressed at threshold $t$. Note that $\text{mAP}(t)$ tends to decrease with increasing threshold $t$. However, for an ideal universal image embedding, it is expected to still have a high mAP even at threshold $\tau(1)$, due to its strong discriminability between true positives and false positives.

In practice, to numerically compute Equation (2), we uniformly sample 11 thresholds and average $\text{mAP}(t)$ over them:

$$t\text{-mAP} = \frac{1}{11} \sum_{t \in \{0, \tau(0.1), \ldots, \tau(1.0)\}} \text{mAP}(t). \tag{3}$$

As can be seen, $t$-mAP takes value from $[0, 1]$, with higher value indicating better performance.

Table 3: The training data used by different image retrieval methods. "Web images" means the training data are sourced from the Internet and typically comprise various 3D objects along with some flat objects. We use the generic term "3D objects" to indicate the training data involve diverse 3D objects, such as 3D landmarks, plants, and animals, etc.

| Method | Training data | Domain | # images | Method | Training data | Domain | # images |
|---|---|---|---|---|---|---|---|
| BoW [8] | - | - | - | BLIP [23] | 129M [23] | 3D objects + web images | 129M |
| FIRe [47] | SfM-120k [39] | 3D landmark | 120K | BLIP2 [22] | 129M [23] | 3D objects + web images | 129M |
| DELG [4] | GLD [30] | 3D landmark | 960K | DINO [5] | ImageNet [9] | 3D objects | 1M |
| CLIP [40] | Proprietary 400M | Web images | 400M | DINOv2 [32] | LVD-142M [32] | 3D objects | 142M |
| SLIP [28] | YFCC15M [43] | Web images | 15M | DiHT [36] | LAION-438M [42] | Web images | 438M |

## 4 Experiments

In this section, we evaluate several representative image retrieval methods on our FORB benchmark. Based on the evaluation results, we also provide a detailed analysis on the behavior of different image embeddings and their intriguing properties.

### 4.1 Baseline Methods

We consider 10 existing image retrieval methods as baselines and investigate their image embedding qualities. According to how the embedding is built, these methods can be categorized into 3 groups.

**Bottom-up** This strategy builds a global image embedding based on local image features. The related methods include: (1) BoW [2]. This method extracts RootSIFT [2] local features from the given image, which are then quantized using a codebook and finally assembled into a sparse feature vector, *i.e.*, image embedding. Since BoW only relies on handcrafted low-level image features, the produced embedding tends to have better generalization ability than learning-based ones that fit to certain domains. (2) FIRe [47], which extracts mid-level image features and then aggregates them in a manner similar to BoW. However, different from BoW, FIRe is deep learning-based and the feature extraction needs to be learned with certain training data, *e.g.*, SfM-120k [39].

**Top-down** Contrary to the bottom-up approach, this strategy learns to extract local image features through image-level supervision on global image embeddings. The local features typically correspond to the convolutional feature maps and are used for feature matching or reranking. In contrast, the global image embeddings are used in the first stage of a retrieval system to efficiently select the most similar images. In our experiment, we consider one representative approach, DELG [4], which jointly extracts deep local features and global image embeddings.

**Top-only** This strategy performs image retrieval with learned global image embeddings directly, without the need of extracting and using local image features. The global image embeddings are typically produced from a deep model that is trained on a large dataset, in a supervised or self- / weakly-supervised manner. In the experiment, we consider the following state-of-the-art methods: (1) CLIP [40]; (2) SLIP [28]; (3) BLIP [23]; (4) BLIP2 [22]; (5) DINO [5]; (6) DINOv2 [32]; (7) DiHT [36]. Note that apart from the design differences, another major distinction among these methods lies in their training data; see Table 3 for more details. In Table 6 we also show the specific neural network model used in each method.

It is worth mentioning that for some top-only methods, their training data may overlap with our FORB benchmark. In particular, we find a few images from FORB are also included in LAION-5B [42], and therefore training data based on the subset of LAION-5B (*e.g.*, LAION-438M [42] and 129M [23]) may also share duplicate images with FORB. In addition, since the training set of CLIP are collected from web, it may overlap with FORB as well. This test set overlap issue has been discussed in previous works [40, 42] and is considered to have little impact on the validity of performance evaluations. In the supplementary material we perform extra experiments on a deduplicated version of FORB and observe the evaluation results closely resemble those from the original FORB (see Section A.4)

Table 4: Comparison of different image retrieval methods on our FORB benchmark. **Bolded** numbers indicate the **best** results. † means the model training data may overlap with FORB and the retrieval accuracy can be interpreted as an "upper bound" performance.

| Method | mAP@5 (%) | | | | $t$-mAP@5 (%) | | | |
|---|---|---|---|---|---|---|---|---|
| | Overall | Easy | Medium | Hard | Overall | Easy | Medium | Hard |
| BoW [2] | 78.44 | 90.38 | 79.78 | 52.65 | 62.49 | 78.29 | 63.61 | 35.00 |
| BoW (+ rerank) [2] | 80.38 | 92.69 | 81.77 | 53.70 | 67.83 | 81.95 | 69.04 | 41.03 |
| FIRe [47] | 88.08 | **98.48** | **90.14** | 56.58 | **77.50** | **90.38** | **79.41** | **44.97** |
| DELG [4] | 48.81 | 79.45 | 48.11 | 24.48 | 34.92 | 65.44 | 33.79 | 15.04 |
| DELG (+ rerank) [4] | 58.74 | 87.96 | 58.43 | 31.91 | 39.47 | 70.64 | 38.45 | 17.74 |
| CLIP† [40] | **89.36** | 98.23 | 90.00 | **73.84** | 67.23 | 87.10 | 67.48 | 44.27 |
| SLIP [28] | 39.01 | 64.45 | 38.58 | 17.22 | 24.43 | 50.27 | 23.42 | 8.07 |
| BLIP† [23] | 74.11 | 94.67 | 74.65 | 47.53 | 49.98 | 81.31 | 49.58 | 21.89 |
| BLIP2† [22] | 81.73 | 94.28 | 82.72 | 58.85 | 57.11 | 81.59 | 57.43 | 28.77 |
| DINO [5] | 55.20 | 85.08 | 55.28 | 23.79 | 42.28 | 74.51 | 41.75 | 14.56 |
| DINOv2 [32] | 68.86 | 92.85 | 69.53 | 37.51 | 48.21 | 72.04 | 48.44 | 21.39 |
| DiHT† [36] | 84.77 | 96.56 | 85.47 | 65.55 | 60.54 | 83.79 | 61.06 | 31.43 |

## 4.2 Implementation

In the experiment, we resize the query and database images to standardize the inputs, ensuring that the longest side is no more than 480 while maintaining the original aspect ratio. For the baseline methods, we implement BoW in Python according to [2], while for the others we adapt their open source implementations to image retrieval task. Specifically, for both BoW and FIRe, we build the codebook using 10k images randomly sampled from the database images. For DELG, we follow its default protocols and extract multi-scale local and global features for both query and database images. For all top-only methods, we produce multi-scale feature representations as well. To be specific, we firstly build an image pyramid by resizing the input image and then center cropping. In our implementation, to strike a balance between accuracy and inference speed, we use 3 scales, $\{\frac{1}{\sqrt{2}}, 1, \sqrt{2}\}$, for query images, and 7 scales [30] for database images. Next, we compute the global image features at each scale and apply $L_2$ normalization to them. Finally, we aggregate all the features by average-pooling, followed by another $L_2$ normalization step. Such multi-scale features mitigate the issue of lacking scale invariance for top-only methods. In practice, we observe much improved accuracy of multi-scale features compared to the single-scale ones.

The source code for all the implementations is available at https://github.com/pxiangwu/FORB/, and licensed under the MIT license.

## 4.3 Evaluation

In Table 4 we report image retrieval accuracy for different methods in terms of mAP@5 and $t$-mAP@5 (see supplementary material for more results). It can be observed that:

(1) Image embeddings built from handcrafted low-level features can be more universal than many learning-based global image descriptors. In particular, while BoW was introduced decades ago and manually designed, it still outperforms DELG and many top-only methods on our FORB benchmark, demonstrating its strong generalization ability. Moreover, from $t$-mAP it can be observed that BoW is better at separating true positives from irrelevant candidates, giving a larger matching score margin.

(2) Mid-level image features are more discriminative than low-level descriptors, and their induced global image embeddings exhibit a superior generalization ability over OOD domains. In Table 4 we investigate one baseline method, *i.e.*, FIRe, which builds embeddings from mid-level features. It can be observed that FIRe overall achieves the best performance among all baselines, with the highest $t$-mAP while giving an mAP on par with CLIP. To extract mid-level features, FIRe needs a model training procedure. Surprisingly, although FIRe was trained on 3D landmark images, it can still work well on 2D flat object domains. This could be because in principle the mid-level features of FIRe are similar to the low-level ones, but they typically cover a larger image region and thus incorporate more semantic information, leading to much improved discriminative ability.

Table 5: Retrieval accuracies on diverse objects. We report overall mAP and $t$-mAP. **Bolded** numbers indicate the **best** results. † means the model training data may overlap with FORB and the retrieval accuracy can be interpreted as an "upper bound" performance.

| | mAP@5 (%) / $t$-mAP@5 (%) | | | | | | | |
|---|---|---|---|---|---|---|---|---|
| Method | Animated Card | Photorealistic Card | Book Cover | Painting | Currency | Logo | Packaged Goods | Movie Poster |
| BoW [2] | 85.93 / 70.42 | 79.82 / 62.98 | 87.92 / 72.68 | 73.33 / 53.64 | 70.79 / 52.64 | 29.98 / 20.20 | 88.57 / 70.93 | 73.40 / 53.19 |
| BoW (+ rerank) [2] | 89.68 / 76.65 | 84.58 / 70.55 | 89.57 / 77.25 | 77.94 / 62.31 | 73.46 / 59.81 | 20.06 / 16.26 | 82.10 / 70.68 | 76.90 / 61.31 |
| FIRe [47] | **93.92 / 83.50** | **95.69 / 85.17** | 90.55 / **80.40** | 88.61 / **78.24** | 81.57 / 69.72 | 42.50 / 33.32 | 92.69 / **81.21** | 85.35 / **71.03** |
| DELG [4] | 53.86 / 43.42 | 43.78 / 24.95 | 58.83 / 39.66 | 29.75 / 16.08 | 65.64 / 47.77 | 13.45 / 7.92 | 69.88 / 46.37 | 42.09 / 25.10 |
| DELG (+ rerank) [4] | 64.95 / 50.42 | 55.63 / 28.38 | 67.91 / 42.50 | 39.83 / 18.24 | 73.94 / 50.93 | 19.17 / 9.91 | 76.23 / 48.32 | 49.80 / 27.01 |
| CLIP† [40] | 91.93 / 72.91 | 74.26 / 54.00 | **99.17** / 71.48 | 93.12 / 63.43 | **87.30 / 73.95** | **85.29 / 53.90** | 98.14 / 79.10 | **86.99** / 53.92 |
| SLIP [28] | 34.15 / 24.84 | 47.51 / 34.30 | 45.50 / 22.34 | 55.93 / 26.71 | 29.74 / 24.20 | 14.92 / 3.25 | 64.12 / 29.64 | 38.28 / 19.52 |
| BLIP† [23] | 64.87 / 51.64 | 74.22 / 58.61 | 93.43 / 55.50 | 79.60 / 45.10 | 68.93 / 50.17 | 82.86 / 29.25 | 96.37 / 51.04 | 69.51 / 32.64 |
| BLIP2† [22] | 78.21 / 64.47 | 78.23 / 57.77 | 96.44 / 61.86 | 84.43 / 42.36 | 78.32 / 55.11 | 80.59 / 31.74 | 97.70 / 63.32 | 73.53 / 33.82 |
| DINO [5] | 52.75 / 41.27 | 80.16 / 63.26 | 48.78 / 33.20 | 65.90 / 51.90 | 53.49 / 41.27 | 6.15 / 3.14 | 77.15 / 56.19 | 55.47 / 41.11 |
| DINOv2 [32] | 70.00 / 45.29 | 87.76 / 76.93 | 68.56 / 37.68 | 79.92 / 57.40 | 65.30 / 55.83 | 6.62 / 1.44 | 92.26 / 68.55 | 65.04 / 35.88 |
| DiHT† [36] | 83.02 / 67.74 | 78.22 / 62.49 | 95.66 / 65.24 | **93.25** / 47.97 | 84.41 / 59.14 | 78.09 / 26.77 | **98.38** / 73.18 | 80.33 / 37.85 |

Table 6: Architectures and inference speeds (seconds / query) of different methods.

| Method | Architecture | Speed | Method | Architecture | Speed |
|---|---|---|---|---|---|
| BoW [2] | - | 0.410 | SLIP [28] | ViT-L/16 | 0.209 |
| BoW (+ rerank) [2] | - | 0.418 | BLIP [23] | ViT-L/16 | 0.211 |
| FIRe [47] | ResNet-50 | **0.124** | BLIP2 [22] | ViT-g/14 + QFormer | 0.341 |
| DELG [4] | ResNet-50 | 0.376 | DINO [5] | ViT-S/8 | 0.177 |
| DELG (+ rerank) [4] | ResNet-50 | 6.015 | DINOv2 [32] | ViT-L/14 | 0.222 |
| CLIP [40] | ViT-L/14@336px | 0.513 | DiHT [36] | ViT-L/14@336px | 0.357 |

(3) For top-only methods, their retrieval accuracies on OOD domains improve with increasing size of model and training data. For example, since the training data of DINO and SLIP are relatively smaller than others, the generalization ability of their image embeddings is inferior to that of CLIP and DiHT, which employ larger model and training set.

(4) Image embeddings based on low- and mid-level features cannot adequately distinguish feature-scarce images. As shown in Table 5, both BoW and FIRe fail to accurately recognize logos, which typically consist of simple patterns and contain sparse features. In contrast, the top-only methods are better at handling logos, probably because they describe images based on their high-level semantics and thus suffer less from the lack of lower level features.

In addition to the retrieval accuracies, we also show the inference speeds of different methods in Table 6. We measure the speed on a machine with 120 GB RAM, an NVIDIA T4 GPU and 32 Intel Xeon CPUs (@2.30GHz). Notably, although CLIP achieves the highest mAP among all the methods, it is not efficient since the feature extraction is computationally expensive. In contrast, FIRe runs at a much faster speed, with a similar mAP and even better $t$-mAP. This further demonstrates the advantages of bottom-up strategy and mid-level features.

## 4.4 Discussion

As shown in Table 4, while being effective on certain object domains, the embeddings from most of the baseline methods are not universal enough to generalize to diverse open-world objects. This affirms the need for the proposed FORB benchmark to further strengthen the research in the generalization ability of image embeddings. In addition, our benchmark results show that even trained with 3D landmark images, embeddings produced by FIRe can still well distinguish images from OOD domains, indicating the great potential of mid-level features in retrieval tasks. In particular, given the advantages and weaknesses of mid-level features, one future direction would be to develop a retrieval method that jointly leverages the mid- and high-level image features, giving image embeddings that share the benefits of both sides.

## 5 Conclusion

We present FORB, a benchmark for flat object retrieval and matching. Essentially FORB supplements existing image retrieval benchmarks, and more importantly, it serves as a test bed for evaluating the generalization abilities of image embeddings on OOD domains. Our experiments on FORB shows

that embeddings based on low- and mid-level image features overall are more universal than those constructed from high-level semantics. Notably, we observe that the mid-level features introduced by FIRe are surprisingly general and give the best overall retrieval performance, even if the model is trained on 3D landmarks. However, despite the overall inferiority, embeddings of high-level semantics are usually more effective for images that contain sparse features. These findings suggest that one potential future direction would be to develop methods that jointly leverage the mid- and high-level image features and combine the strengths of both.

**Limitations and future work.** In our experiment, we compare baselines which have different model sizes and are trained on various datasets. As a result, the comparisons among these methods and their corresponding embeddings could be unfair to some extent. In addition, our FORB benchmark currently only considers distractors from the same domain as the index images. To improve the diversity and challenges of our benchmark, in the future we plan to collect more distractors from other domains. In addition, to further enrich the OOD queries, we also plan to curate queries beyond the domains of index images, a practice which is similar to GLDv2 [49]. In this way we can better measure the matching score margins of different methods with $t$-mAP. Despite these limitations, our benchmark still serves as a supportive dataset for further research in the task of image retrieval. We hope our work would facilitate the understanding of different image embeddings and promote the design of new methods.

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
