# FORB: A Flat Object Retrieval Benchmark for Universal Image Embedding
## - Supplementary Material -

**Pengxiang Wu, Siman Wang, Kevin Dela Rosa, Derek Hao Hu**
Snap Inc.
{pwu,swang7,kevin.delarosa,hao.hu}@snap.com

## A  Appendix

### A.1  More Query Images with Different Difficulties

In Figures 1 - 8 we provide additional sample images from our FORB benchmark. For each flat object type, we showcase query images of different difficulties and their corresponding index image. Query images with difficulty "hard" overall present the greatest retrieval challenge, due to truncation, occlusion, perspective transformation, and the distraction of background. Note that in our benchmark, for objects with the same pattern but in different colors, we consider them equivalent and matched; see the easy query and index images in Figure 6.

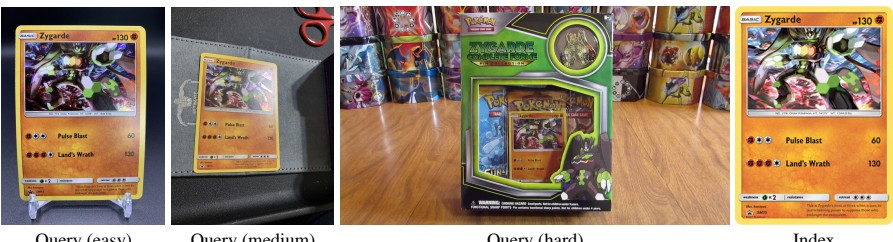

| Query (easy) | Query (medium) | Query (hard) | Index |
|---|---|---|---|

Figure 1: Example images of **animated trading card**. In the query image with "hard" difficulty, the target object only occupies a small area, making it non-trivial to recognize due to the distraction of background. In contrast, in "easy" and "medium" queries, the target object occupies a larger area.

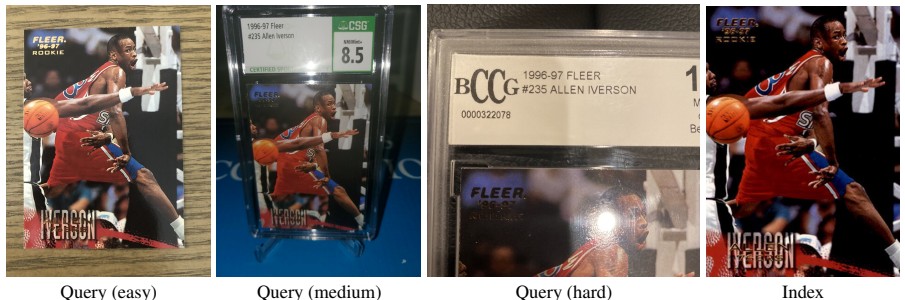

| Query (easy) | Query (medium) | Query (hard) | Index |
|---|---|---|---|

Figure 2: Example images of **photorealistic trading card**. In the query image with "hard" difficulty, the target trading card is severely truncated and thus difficult to recognize.

37th Conference on Neural Information Processing Systems (NeurIPS 2023) Track on Datasets and Benchmarks.

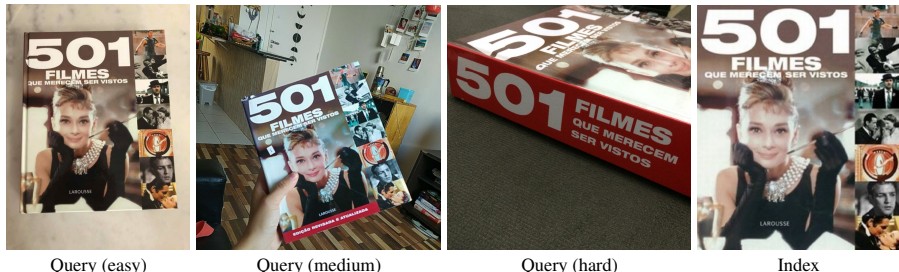

Figure 3: Example images of **book cover**. In the query image with "hard" difficulty, the target book cover is truncated and under a large perspective transformation.

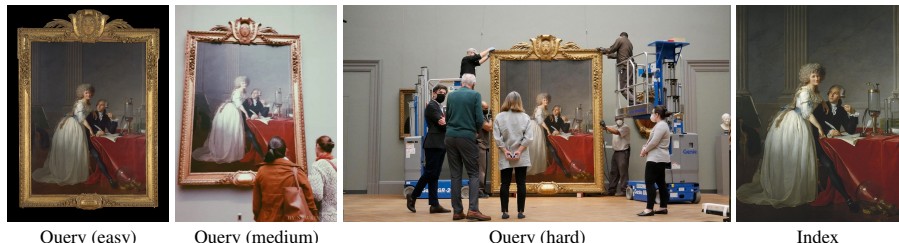

Figure 4: Example images of **painting**. In the query image with "hard" difficulty, the target painting is occluded and occupies a small area.

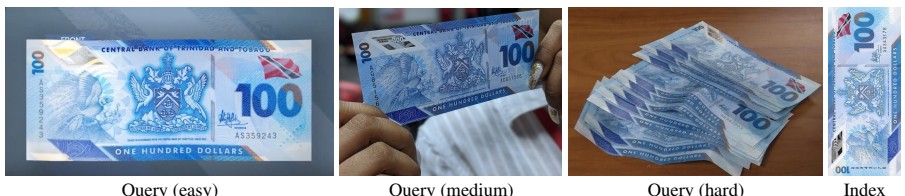

Figure 5: Example images of **currency**. In the query image with "hard" difficulty, the target currency is blurry and only occupies a small area.

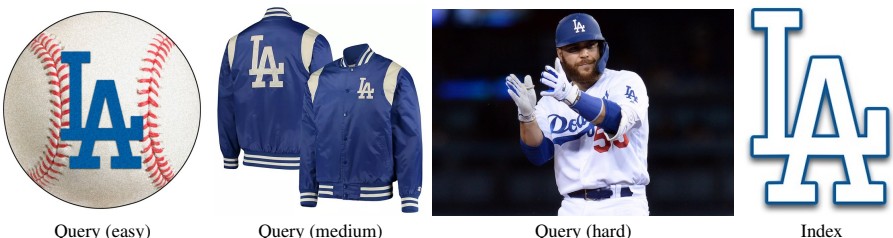

Figure 6: Example images of **logo**. In the query image with "hard" difficulty, the target logo only occupies a small area, with much distraction from the background.

## A.2 Distractor Images

In Figure 9, we illustrate different distractor images that bear similarities to the index images in various aspects. For example, in Figures 9(a)(b) both index and distractor images share very similar contents and textures. In Figure 9(c), although the index and distractor images have different contents, they share similar styles. In Figure 9(d), both logos are similar in shapes. In Figure 9(e), the index and distractor images contain the same texts, which would pose challenges for text-sensitive methods such as CLIP [10]. In Figure 9(f), the index and distractor images refer to similar products of the same brand. In Figure 9(g), both index and distractor images are about the same movie and thus share the same semantics. This would pose challenges for top-only methods since their embeddings capture more about high-level semantics. Similarly, in Figures 9(h)(i) both index and distractor images refer to the same person or object.

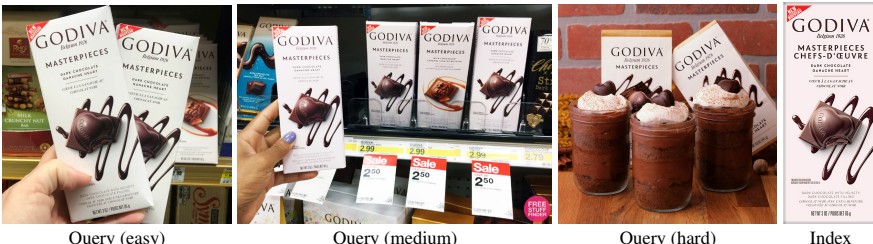

| Query (easy) | Query (medium) | Query (hard) | Index |

Figure 7: Example images of **packaged goods**. In the query image with "hard" difficulty, the target packaged goods is heavily occluded.

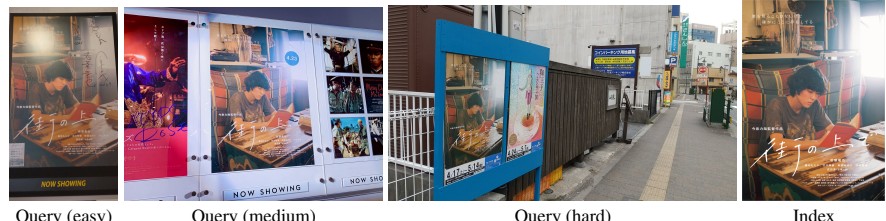

| Query (easy) | Query (medium) | Query (hard) | Index |

Figure 8: Example images of **movie poster**. In the query image with "hard" difficulty, the target movie poster only occupies a small area and is under perspective transformation.

### A.3 Additional Evaluation Results on FORB

In Tables 1 and 2 we show additional benchmark results on FORB in terms of mAP@1 and $t$-mAP@1. Note that mAP@1 is essentially equivalent to Recall@1. We observe that FIRe overall achieves the best performance. This demonstrates its superior generalization ability on OOD domains.

### A.4 Evaluation Results on FORB Subset

The baselines considered in our experiments are trained on various datasets, some of which overlap with our FORB dataset. In particular, we find a few images from FORB are also included in LAION-5B [11], and as a result, training sets based on the subset of LAION-5B (*e.g.*, LAION-438M [11] and 129M [6]) may also share duplicate images with FORB. To make FORB a real OOD query set and evaluate the generalization abilities of baselines trained on LAION-based datasets, we remove duplicate images shared with LAION-5B from FORB and compare baselines on the reduced FORB dataset. In total the number of duplicate images is small: only 5.91% query images and 19.93% database images are included in LAION-5B. In Tables 3-6 we report the evaluation results on the reduced FORB benchmark. Note that since the training data used by CLIP [10] are not publicly available, it is difficult to determine if FORB overlaps with it. Therefore, the performance of CLIP in Tables 3-6 can be considered as an "upper bound".

In Tables 3-6, we observe that the results on reduced FORB are very similar to those from the original FORB dataset. This shows that duplicate images only have little impact on the evaluations.

### A.5 Index File Size

In Table 7 we provide the size of index files for each baseline method. In our implementation, we do not use approximate nearest neighbors (ANN) search for top-only methods, but instead simply use matrix multiplication to compute cosine similarities between image embeddings and sort the database candidates accordingly. In fact, we observe that naive GPU-based matrix multiplication already achieves decent search latency. As can be seen in Table 7, compared to top-only methods, both the bottom-up and top-down methods have a relatively larger size of index file. In particular, the index file size of BoW and DELG is orders of magnitude larger than others, making it difficult to scale up to larger database in practice. In contrast, FIRe requires a much smaller index file whose size is similar to those of top-only methods. Given the superior inference speed and retrieval accuracy, FIRe overall outperforms other methods and is suitable for deployment in real applications. Based on

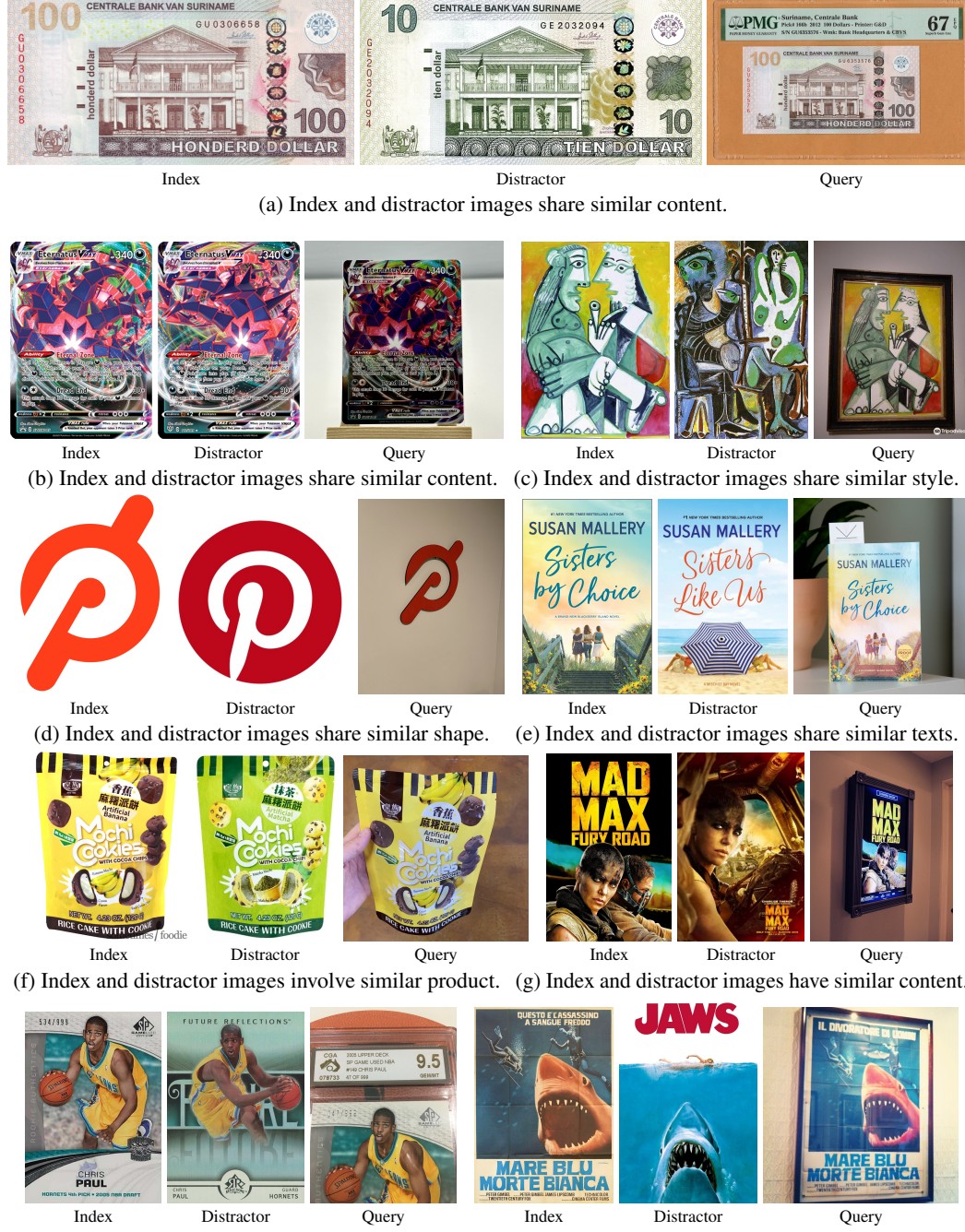

(a) Index and distractor images share similar content.

(b) Index and distractor images share similar content. (c) Index and distractor images share similar style.

(d) Index and distractor images share similar shape. (e) Index and distractor images share similar texts.

(f) Index and distractor images involve similar product. (g) Index and distractor images have similar content.

(h) Index and distractor images refer to the same person. (i) Index and distractor images contain same object.

Figure 9: Examples of index and distractor images. We show distractor images that are similar to the index images in various aspects.

the aforementioned observations, we believe one promising future direction is to develop methods that combine the benefits of FIRe and CLIP-like approaches.

## A.6   Data Format

The query and database images are available in JPEG format, which can be easily read by many existing (Python) libraries. It is worth mentioning that the original images on the Internet are not

Table 1: Comparison of different image retrieval methods on our FORB benchmark. We measure the performance in terms of mAP@1 and $t$-mAP@1. **Bolded** numbers indicate the **best** results. † means the model training data may overlap with FORB and the retrieval accuracy can be interpreted as an "upper bound" performance.

| Method | mAP@1 (%) | | | | $t$-mAP@1 (%) | | | |
|---|---|---|---|---|---|---|---|---|
| | Overall | Easy | Medium | Hard | Overall | Easy | Medium | Hard |
| BoW [1] | 74.69 | 87.35 | 76.07 | 47.79 | 60.29 | 76.12 | 61.43 | 32.60 |
| BoW (+ rerank) [1] | 78.88 | 91.05 | 80.31 | 51.95 | 66.65 | 80.52 | 67.90 | 39.79 |
| FIRe [12] | **86.57** | **98.11** | **88.67** | 53.52 | **76.70** | **90.14** | **78.59** | **43.76** |
| DELG [2] | 44.88 | 75.73 | 44.07 | 21.42 | 32.91 | 63.02 | 31.74 | 13.84 |
| DELG (+ rerank) [2] | 58.33 | 87.82 | 57.99 | 31.46 | 39.26 | 70.52 | 38.24 | 17.55 |
| CLIP† [10] | 84.35 | 96.58 | 84.97 | **65.64** | 64.24 | 85.72 | 64.42 | 40.43 |
| SLIP [7] | 32.61 | 58.11 | 31.98 | 12.81 | 21.09 | 45.88 | 20.02 | 6.47 |
| BLIP† [6] | 68.04 | 91.56 | 68.54 | 38.85 | 46.74 | 79.02 | 46.23 | 18.84 |
| BLIP2† [5] | 75.28 | 90.37 | 76.30 | 49.50 | 53.57 | 78.56 | 53.84 | 25.25 |
| DINO [3] | 51.79 | 83.05 | 51.64 | 21.29 | 40.62 | 73.11 | 39.99 | 13.60 |
| DINOv2 [8] | 65.19 | 90.98 | 65.74 | 33.25 | 46.35 | 70.93 | 46.50 | 19.69 |
| DiHT† [9] | 79.70 | 93.94 | 80.41 | 57.92 | 57.90 | 81.86 | 58.33 | 29.01 |

Table 2: Retrieval accuracies on diverse objects. We report overall mAP@1 and $t$-mAP@1. **Bolded** numbers indicate the **best** results. † means the model training data may overlap with FORB and the retrieval accuracy can be interpreted as an "upper bound" performance.

| Method | mAP@1 (%) / $t$-mAP@1 (%) | | | | | | | |
|---|---|---|---|---|---|---|---|---|
| | Animated Card | Photorealistic Card | Book Cover | Painting | Currency | Logo | Packaged Goods | Movie Poster |
| BoW [1] | 82.07 / 67.92 | 75.95 / 60.72 | 85.90 / 71.45 | 69.94 / 52.21 | 63.46 / 48.50 | 25.73 / 18.01 | 86.12 / 69.58 | 70.31 / 52.04 |
| BoW (+ rerank) [1] | 87.45 / 74.88 | 83.81 / 69.99 | 88.98 / 76.78 | 77.53 / 61.97 | 70.71 / 57.72 | 18.94 / 15.39 | 81.13 / 69.94 | 76.37 / 60.90 |
| FIRe [12] | **93.03** / **82.99** | **94.56** / **84.57** | 89.53 / **79.89** | 86.64 / **77.19** | 79.82 / 68.99 | 37.19 / 30.31 | 91.37 / **80.64** | **83.20** / **70.21** |
| DELG [2] | 49.44 / 40.46 | 39.37 / 23.45 | 55.51 / 38.42 | 26.32 / 15.07 | 61.87 / 46.03 | 10.75 / 6.87 | 66.25 / 45.00 | 39.65 / 24.34 |
| DELG (+ rerank) [2] | 64.68 / 50.24 | 55.42 / 28.31 | 67.62 / 42.33 | 39.37 / 18.11 | 73.48 / 50.71 | 17.69 / 9.25 | 75.50 / 47.95 | 49.61 / 26.97 |
| CLIP† [10] | 88.17 / 70.45 | 66.58 / 49.27 | **98.63** / 71.18 | 90.18 / 62.27 | **82.06** / **69.92** | **68.72** / **45.21** | 96.75 / 78.14 | **83.20** / 52.57 |
| SLIP [7] | 28.37 / 21.13 | 39.14 / 29.25 | 40.31 / 20.65 | 47.57 / 24.25 | 23.35 / 19.05 | 11.58 / 2.69 | 53.50 / 26.34 | 32.81 / 17.63 |
| BLIP† [6] | 59.07 / 47.65 | 67.63 / 54.20 | 92.13 / 55.18 | 74.39 / 43.40 | 62.40 / 46.49 | 66.37 / 24.77 | 96.25 / 62.70 | 65.04 / 31.43 |
| BLIP2† [5] | 71.62 / 59.69 | 71.15 / 53.58 | 95.35 / 61.61 | 80.06 / 41.47 | 70.98 / 51.32 | 64.26 / 26.78 | 96.25 / 62.70 | 68.36 / 32.33 |
| DINO [3] | 48.86 / 39.10 | 77.37 / 61.95 | 45.17 / 32.00 | 62.85 / 50.62 | 50.13 / 39.59 | 4.70 / 2.75 | 72.25 / 53.99 | 52.73 / 39.84 |
| DINOv2 [8] | 65.46 / 42.88 | 84.82 / 74.67 | 65.71 / 37.06 | 76.52 / 56.31 | 61.21 / 52.59 | 4.47 / 1.18 | 89.62 / 67.13 | 61.33 / 35.07 |
| DiHT† [9] | 78.36 / 64.58 | 72.15 / 58.53 | 94.32 / 64.64 | **90.79** / 47.41 | 78.23 / 56.19 | 63.68 / 23.02 | **97.38** / 72.63 | 75.78 / 36.77 |

necessarily in JPEG format. We standardize them to JPEG format with a script, which is also available on our GitHub repository. For metadata, including annotations and lists of images, they are organized in newline delimited JSON files, which can be loaded with (Python) json library.

## A.7 Licensing and Maintenance Schedule

The dataset link and downloaders of FORB are maintained by the authors on GitHub. In particular, the data (including images and metadata) are accessible under the CC BY-NC-SA license. All the supporting code is available on the same GitHub repository, licensed under the MIT license. Any issues or discussions regarding technical or other concerns can be submitted to the GitHub repository, and the authors will reply as soon as possible. Community forks and pull requests will be welcome and reviewed by the repository maintainers.

Our FORB benchmark is a growing project. In the future we expect to include more object types as well as to increase the quantities of both query and database images. New versions of FORB dataset will be shared and announced on the GitHub page (https://github.com/pxiangwu/FORB/). We maintain the history of versions and will provide the download link to each version. Finally, we expect to also include new emerging baseline methods to establish up-to-date benchmark results.

## A.8 Author Statement

In accordance with the CC BY-NC-SA and MIT license, the authors bear all responsibility in case of violation of rights. The descriptions made in the paper and its supplementary material are accurate and agreed upon by all authors.

Table 3: Comparison of different image retrieval methods on reduced FORB dataset. We measure the performance in terms of mAP@5 and $t$-mAP@5. **Bolded** numbers indicate the **best** results. † means the model training data may overlap with FORB and the retrieval accuracy can be interpreted as an "upper bound" performance.

| Method | mAP@5 (%) | | | | $t$-mAP@5 (%) | | | |
|---|---|---|---|---|---|---|---|---|
| | Overall | Easy | Medium | Hard | Overall | Easy | Medium | Hard |
| BoW [1] | 81.20 | 92.90 | 82.00 | 57.92 | 67.06 | 82.58 | 67.68 | 41.35 |
| BoW (+ rerank) [1] | 83.35 | 95.18 | 84.32 | 58.03 | 70.43 | 84.60 | 71.26 | 43.93 |
| FIRe [12] | **90.22** | 98.18 | **91.97** | 60.47 | **79.32** | **89.83** | **81.00** | **47.34** |
| DELG [2] | 51.57 | 83.22 | 50.43 | 26.96 | 36.64 | 69.18 | 35.01 | 16.69 |
| DELG (+ rerank) [2] | 62.31 | 91.58 | 61.63 | 35.35 | 41.60 | 74.72 | 40.07 | 19.94 |
| CLIP† [10] | 89.59 | **98.55** | 90.14 | **72.51** | 67.64 | 87.03 | 67.67 | 44.10 |
| SLIP [7] | 41.43 | 66.33 | 40.89 | 18.00 | 25.92 | 51.59 | 24.78 | 8.59 |
| BLIP [6] | 74.31 | 95.00 | 74.80 | 44.04 | 50.31 | 81.17 | 49.73 | 20.21 |
| BLIP2 [5] | 82.62 | 94.94 | 83.52 | 57.58 | 58.29 | 82.12 | 58.40 | 28.63 |
| DINO [3] | 58.36 | 85.61 | 58.45 | 24.85 | 44.56 | 74.51 | 44.03 | 14.97 |
| DINOv2 [8] | 72.27 | 93.88 | 72.89 | 39.41 | 49.90 | 70.27 | 50.23 | 21.75 |
| DiHT [9] | 85.48 | 97.39 | 85.98 | 65.50 | 62.14 | 84.35 | 62.50 | 31.41 |

Table 4: Comparison of different image retrieval methods on reduced FORB dataset. We measure the performance in terms of mAP@1 and $t$-mAP@1. **Bolded** numbers indicate the **best** results. † means the model training data may overlap with FORB and the retrieval accuracy can be interpreted as an "upper bound" performance.

| Method | mAP@1 (%) | | | | $t$-mAP@1 (%) | | | |
|---|---|---|---|---|---|---|---|---|
| | Overall | Easy | Medium | Hard | Overall | Easy | Medium | Hard |
| BoW [1] | 77.69 | 90.81 | 78.45 | 53.29 | 65.53 | 81.20 | 66.15 | 39.68 |
| BoW (+ rerank) [1] | 82.04 | 94.09 | 83.02 | 56.36 | 69.40 | 83.69 | 70.23 | 42.76 |
| FIRe [12] | **88.81** | **98.01** | **90.60** | 57.04 | **78.53** | **89.69** | **80.19** | **45.98** |
| DELG [2] | 47.36 | 79.05 | 46.10 | 24.20 | 34.46 | 66.35 | 32.81 | 15.59 |
| DELG (+ rerank) [2] | 61.90 | 91.45 | 61.18 | 35.07 | 41.39 | 74.60 | 39.84 | 19.80 |
| CLIP† [10] | 85.40 | 97.62 | 85.90 | **65.08** | 65.13 | 86.26 | 65.07 | 40.62 |
| SLIP [7] | 34.78 | 60.60 | 34.00 | 13.02 | 22.45 | 47.65 | 21.22 | 6.67 |
| BLIP [6] | 69.03 | 92.74 | 69.38 | 36.75 | 47.38 | 79.50 | 46.66 | 17.40 |
| BLIP2 [5] | 76.91 | 91.97 | 77.73 | 49.54 | 55.03 | 79.76 | 55.05 | 25.43 |
| DINO [3] | 54.87 | 83.48 | 54.74 | 22.36 | 42.91 | 73.08 | 42.29 | 14.16 |
| DINOv2 [8] | 68.64 | 92.35 | 69.10 | 35.07 | 48.08 | 69.54 | 48.29 | 20.10 |
| DiHT [9] | 81.05 | 95.69 | 81.43 | 59.11 | 59.72 | 83.06 | 59.95 | 29.34 |

Table 5: Object retrieval accuracies on reduced FORB dataset. We report overall mAP@5 and $t$-mAP@5. **Bolded** numbers indicate the **best** results. † means the model training data may overlap with FORB and the retrieval accuracy can be interpreted as an "upper bound" performance.

| Method | mAP@5 (%) / $t$-mAP@5 (%) | | | | | | | |
|---|---|---|---|---|---|---|---|---|
| | Animated Card | Photorealistic Card | Book Cover | Painting | Currency | Logo | Packaged Goods | Movie Poster |
| BoW [1] | 86.57 / 73.77 | 80.09 / 63.65 | 89.48 / 76.53 | 68.78 / 55.51 | 74.92 / 59.27 | 30.22 / 15.83 | 89.36 / 73.45 | 72.81 / 55.99 |
| BoW (+ rerank) [1] | 89.75 / 76.51 | 85.22 / 70.63 | 90.99 / 79.08 | 72.46 / 58.85 | 76.66 / 62.60 | 20.18 / 16.09 | 84.52 / 73.16 | 74.70 / 59.63 |
| FIRe [12] | **93.67 / 83.15** | **96.02 / 85.16** | 91.04 / **80.53** | 91.59 / **80.11** | 82.37 / 69.69 | 41.58 / 32.22 | 93.25 / **81.43** | 85.26 / **70.60** |
| DELG [2] | 54.92 / 43.82 | 44.54 / 24.91 | 60.37 / 39.59 | 35.77 / 18.60 | 65.88 / 46.42 | 12.46 / 7.85 | 71.32 / 48.02 | 40.58 / 23.56 |
| DELG (+ rerank) [2] | 66.61 / 51.06 | 56.55 / 28.27 | 70.17 / 42.61 | 46.96 / 21.55 | 74.33 / 49.27 | 18.66 / 10.04 | 77.55 / 49.96 | 50.84 / 25.98 |
| CLIP† [10] | 92.78 / 73.34 | 75.98 / 55.09 | **99.38** / 71.89 | 92.91 / 62.60 | **89.74 / 74.85** | **86.57 / 52.03** | 98.42 / 78.95 | **87.42** / 51.99 |
| SLIP [7] | 37.03 / 26.20 | 48.21 / 33.57 | 46.90 / 21.46 | 67.82 / 28.96 | 32.85 / 26.74 | 16.27 / 3.35 | 65.99 / 28.48 | 39.02 / 16.75 |
| BLIP [6] | 66.55 / 51.71 | 74.98 / 57.87 | 94.05 / 52.80 | 83.22 / 43.47 | 72.98 / 51.29 | 83.33 / 25.35 | 97.18 / 47.54 | 68.69 / 28.02 |
| BLIP2 [5] | 79.64 / 64.86 | 80.00 / 58.24 | 96.66 / 59.46 | 88.18 / 40.03 | 82.22 / 55.48 | 82.56 / 27.26 | 97.98 / 59.97 | 73.71 / 29.30 |
| DINO [3] | 54.57 / 42.35 | 80.09 / 62.91 | 49.40 / 33.36 | 69.49 / 54.88 | 54.38 / 41.49 | 5.96 / 2.89 | 77.50 / 56.48 | 52.29 / 38.04 |
| DINOv2 [8] | 71.75 / 44.66 | 88.51 / 77.37 | 67.69 / 35.03 | 81.59 / 58.65 | 67.44 / 57.02 | 7.34 / 1.55 | 92.26 / 66.26 | 62.27 / 31.68 |
| DiHT [9] | 84.56 / 67.97 | 80.11 / 63.44 | 96.21 / 63.69 | **94.04** / 45.86 | 87.69 / 59.59 | 78.65 / 22.42 | **98.88** / 71.26 | 80.65 / 33.23 |

# References

[1] Relja Arandjelović and Andrew Zisserman. Three things everyone should know to improve object retrieval. In *Proceedings of the IEEE conference on computer vision and pattern recognition*, pages 2911–2918. IEEE, 2012.

Table 6: Object retrieval accuracies on reduced FORB dataset. We report overall mAP@1 and $t$-mAP@1. † means the model training data may overlap with FORB and the retrieval accuracy can be interpreted as an "upper bound" performance. **Bolded** numbers indicate the **best** results.

| Method | mAP@1 (%) / $t$-mAP@1 (%) | | | | | | | |
| --- | --- | --- | --- | --- | --- | --- | --- | --- |
| | Animated Card | Photorealistic Card | Book Cover | Painting | Currency | Logo | Packaged Goods | Movie Poster |
| BoW [1] | 83.14 / 71.69 | 75.62 / 62.18 | 87.51 / 76.26 | 67.15 / 55.51 | 69.64 / 56.92 | 26.33 / 15.17 | 86.90 / 72.88 | 69.92 / 55.48 |
| BoW (+ rerank) [1] | 87.75 / 74.93 | 84.54 / 70.15 | 90.93 / 79.02 | 72.46 / 58.85 | 74.33 / 60.82 | 19.51 / 15.56 | 83.82 / 72.57 | 74.09 / 59.20 |
| FIRe [12] | **92.43 / 82.37** | **95.19 / 84.66** | 89.87 / **80.00** | 89.86 / **79.23** | 80.80 / 69.20 | 36.71 / 29.40 | 91.91 / **80.78** | 83.01 / **69.69** |
| DELG [2] | 50.16 / 40.67 | 40.00 / 23.45 | 57.36 / 38.57 | 31.88 / 17.44 | 62.50 / 44.95 | 9.81 / 6.77 | 67.63 / 46.58 | 37.88 / 22.69 |
| DELG (+ rerank) [2] | 66.26 / 50.83 | 56.32 / 28.21 | 69.96 / 42.51 | 46.38 / 21.48 | 73.88 / 48.99 | 17.13 / 9.37 | 76.69 / 49.52 | 50.70 / 25.96 |
| CLIP† [10] | 89.57 / 71.29 | 68.65 / 50.63 | **99.06** / 71.67 | 89.86 / 61.57 | **84.60 / 71.04** | **75.91 / 46.76** | 97.11 / 78.03 | **83.57** / 50.70 |
| SLIP [7] | 30.76 / 22.36 | 40.16 / 28.92 | 41.81 / 20.03 | 59.90 / 26.88 | 26.34 / 21.63 | 12.22 / 2.65 | 55.30 / 25.64 | 34.26 / 15.55 |
| BLIP [6] | 60.97 / 48.12 | 68.49 / 53.77 | 92.93 / 52.59 | 79.23 / 42.38 | 66.96 / 48.36 | 72.60 / 22.94 | 95.76 / 47.10 | 65.18 / 27.27 |
| BLIP2 [5] | 73.31 / 60.43 | 73.68 / 54.69 | 95.64 / 59.31 | 84.06 / 39.13 | 75.00 / 52.17 | 71.32 / 24.59 | 96.72 / 59.43 | 69.36 / 28.34 |
| DINO [3] | 50.61 / 40.26 | 77.51 / 61.76 | 45.82 / 32.13 | 67.15 / 53.93 | 50.45 / 39.87 | 4.41 / 2.43 | 73.03 / 54.55 | 48.75 / 36.44 |
| DINOv2 [8] | 67.40 / 42.53 | 85.78 / 75.26 | 64.66 / 34.52 | 78.26 / 57.62 | 62.95 / 53.59 | 5.15 / 1.31 | 89.40 / 64.81 | 59.05 / 31.05 |
| DiHT [9] | 80.25 / 65.21 | 74.38 / 59.78 | 95.29 / 63.37 | **91.79** / 45.28 | 82.14 / 57.16 | 68.05 / 20.11 | **98.07** / 70.82 | 76.04 / 32.36 |

Table 7: The index file sizes of different image retrieval methods.

| Method | BoW (+ rerank) [4] | FIRe [12] | DELG (+ rerank) [2] | CLIP [10] | SLIP [7] |
| --- | --- | --- | --- | --- | --- |
| Index file size | 22.141 GB | 0.563 GB | 15.222 GB | 0.082 GB | 0.107 GB |
| Method | BLIP [6] | BLIP2 [5] | DINO [3] | DINOv2 [8] | DiHT [9] |
| Index file size | 0.214 GB | **0.054 GB** | 0.082 GB | 0.214 GB | 0.160 GB |

[2] Bingyi Cao, Andre Araujo, and Jack Sim. Unifying deep local and global features for image search. In *European conference on computer vision*, pages 726–743. Springer, 2020.

[3] Mathilde Caron, Hugo Touvron, Ishan Misra, Hervé Jégou, Julien Mairal, Piotr Bojanowski, and Armand Joulin. Emerging properties in self-supervised vision transformers. In *Proceedings of the IEEE/CVF international conference on computer vision*, pages 9650–9660, 2021.

[4] Gabriella Csurka, Christopher Dance, Lixin Fan, Jutta Willamowski, and Cédric Bray. Visual categorization with bags of keypoints. In *Workshop on statistical learning in computer vision, ECCV*, volume 1, pages 1–2. Prague, 2004.

[5] Junnan Li, Dongxu Li, Silvio Savarese, and Steven Hoi. Blip-2: Bootstrapping language-image pre-training with frozen image encoders and large language models. *International conference on machine learning*, 2023.

[6] Junnan Li, Dongxu Li, Caiming Xiong, and Steven Hoi. Blip: Bootstrapping language-image pre-training for unified vision-language understanding and generation. In *International conference on machine learning*, pages 12888–12900. PMLR, 2022.

[7] Norman Mu, Alexander Kirillov, David Wagner, and Saining Xie. Slip: Self-supervision meets language-image pre-training. In *European conference on computer vision*, pages 529–544. Springer, 2022.

[8] Maxime Oquab, Timothée Darcet, Théo Moutakanni, Huy Vo, Marc Szafraniec, Vasil Khalidov, Pierre Fernandez, Daniel Haziza, Francisco Massa, Alaaeldin El-Nouby, et al. Dinov2: Learning robust visual features without supervision. *arXiv preprint arXiv:2304.07193*, 2023.

[9] Filip Radenovic, Abhimanyu Dubey, Abhishek Kadian, Todor Mihaylov, Simon Vandenhende, Yash Patel, Yi Wen, Vignesh Ramanathan, and Dhruv Mahajan. Filtering, distillation, and hard negatives for vision-language pre-training. *Proceedings of the IEEE/CVF conference on computer vision and pattern recognition*, 2023.

[10] Alec Radford, Jong Wook Kim, Chris Hallacy, Aditya Ramesh, Gabriel Goh, Sandhini Agarwal, Girish Sastry, Amanda Askell, Pamela Mishkin, Jack Clark, et al. Learning transferable visual models from natural language supervision. In *International conference on machine learning*, pages 8748–8763. PMLR, 2021.

[11] Christoph Schuhmann, Romain Beaumont, Richard Vencu, Cade Gordon, Ross Wightman, Mehdi Cherti, Theo Coombes, Aarush Katta, Clayton Mullis, Mitchell Wortsman, et al. Laion-5b: An open large-scale dataset for training next generation image-text models. *arXiv preprint arXiv:2210.08402*, 2022.

[12] Philippe Weinzaepfel, Thomas Lucas, Diane Larlus, and Yannis Kalantidis. Learning super-features for image retrieval. *International conference on learning representations*, 2022.