# OpenReview forum: "FORB: A Flat Object Retrieval Benchmark for Universal Image Embedding"
_NeurIPS.cc/2023/Track/Datasets_and_Benchmarks — NeurIPS 2023 Datasets and Benchmarks Poster_

### Official Review · Reviewer_XRLd · 2023-07-18
**review with rating-6**

**Rating:** 6
**Confidence:** 4
**Clarity:** The paper is written well.

**Strengths:**

- The introduced dataset is a good asset to the community.
- The proposed metric t-mAP makes sense.
- The study of baseline methods is extensive and good.
- The paper reads well.


**Additional Feedback:**

Please see above boxes to improve the submission.

**Correctness:**

Authors should carefully draw conclusions and make statements because compared methods are trained on different training data, and it is hard to conclude one type of methods (e.g., mid-level features) is better than others based on their performance.

**Documentation:**

Documentation is good with a link to download the dataset, sufficient details in data collection, and plans to maintain this dataset.

**Ethics:**

The paper doesn't discuss potential ethical issues. It might be good to discuss potential unethical contents included in the new dataset.

**Limitations:**

The paper discusses some limitations already, which make sense.


**Opportunities For Improvement:**

- Line9, Table 2 and Table 3: The compared methods are trained on some datasets, different from the introduced FORB dataset, hence out-of-distribution. However, their training datasets are all different, and directly comparing their performance on FORB is unfair and hard to draw conclusions. It might be possible one training set is better than the other and so leads the corresponding method to better performance. Also because of this, authors might want to augment Table 2 with more information about the training data, e.g., the number of training images, etc. Authors should carefully handle this issue and make proper statements and conclusions.

- While FORB seems like an out-of-distribution dataset compared to training data in Table 2, it is hard to guarantee FORB images are not included in these datasets such as LAION-438M. Authors should clarify and rework Table 2 with further details (e.g., what domain each training dataset is).

- Line13: It is unclear what heterogeneity means. Authors should clarify.

- The paper should make a table to compare other existing datasets for image retrieval. This puts the introduced FORB in the context of relevant literature and helps readers understand the significance of the dataset.

**Relation To Prior Work:**

The paper can do better by making a table to comparing against existing image retrieval benchmarks.

**Summary And Contributions:**

The paper introduces a new dataset for benchmarking visual search methods on flat images with diverse patterns, termed flat object retrieval benchmark (FORB). The benchmark serves as a testbed for assessing the image embedding quality and has no training data. It studies multiple baseline methods and shows that some of them generalizes well to this new dataset. The dataset is valuable. The writing is good. Experiments are good.

---

> ### Author Response · Authors · 2023-08-21
> **Thanks for your constructive feedback**
>
> **Q1: It is unfair to compare the performance of different methods on FORB because these methods are trained on different training sets. Therefore it is hard to draw conclusions. It might be possible one training set is better than the other and so leads the corresponding method to better performance.**
>
> **A:** One major goal of FORB is to measure the generalization ability of embedding models through image retrieval task. With this benchmark, we seek to understand the effects of image features from different levels on the embedding quality, thereby shedding light on future development of embedding models. In other words, we aim to figure out which factors or designs would positively contribute to the embedding quality. To this end, in our experiments we intentionally compare different methods that are trained on different training data, thereby introducing various controlled factors into the experiments.
>
> In fact, from Tables 2 and 3 it can be observed that: (1) Image embeddings built from handcrafted low-level features can be more universal than many learning-based global image descriptors. (2) Mid-level image features are more discriminative than low-level descriptors, and their induced global image embeddings exhibit a superior generalization ability over OOD domains. (3) For top-only methods, their retrieval accuracies on OOD domains improve with increasing size of model and training data. In particular, larger training data usually contain more diverse image patterns and thus would enhance the robustness of learned embeddings. (4) Image embeddings based on low- and mid-level features cannot adequately distinguish feature-scarce images. Please see Section 4.3 for more details.
>
> Therefore, in the above sense it is reasonable to compare the performance of different methods on FORB, even though their training sets are different. Our conclusions are drawn based on such specific experimental settings.
>
> **Q2: Authors might want to augment Table 2 with more information about the training data, e.g., the number of training images, etc.**
>
> **A:** Thank you for the suggestion. We have augmented Table 2 to include more information about the training data. Also note that the number in the training dataset name roughly indicates the number of training images. For example, the “438M” in LAION-438M means there are about 438 million training images.

---

> > ### Author Response · Authors · 2023-08-21
> > **Thanks for your constructive feedback**
> >
> > **Q3: It is hard to guarantee FORB images are not included in these datasets such as LAION-438M. Authors should clarify and rework Table 2 with further details (e.g., what domain each training dataset is).**
> >
> > **A:** Thank you for the suggestion. For LAION-438M, it is a subset of LAION-5B, which contains 5 billion images. However, since the authors did not release the metadata, we are unclear about which images from LAION-5B are included in LAION-438M. Therefore we estimate an upperbound duplication rate by comparing FORB against LAION-5B. The estimation is performed with the following steps. (1) Extract the CLIP embedding for each FORB image. We use ViT-L/14 model. (2) Extract the CLIP embedding for each LAION-5B image using ViT-L/14 and build Faiss approximate nearest neighbour (ANN) index. We use off-the-shelf ANN index provided by LAION-5B. (3) Query each FORB image against LAION-5B and obtain similar images. (4) Manually check if the similar images from LAION-5B are duplicates or not. Eventually we find the duplication rate is small: only 5.91% query images and 19.93% database images are included in LAION-5B. Therefore, for LAION-438M we expect the duplication rate would be even smaller.
> >
> > Similarly, since the training data used by BLIP and BLIP2 contain LAION-100M, which is a 100 million subset of LAION-5B, they may also share duplicate images with FORB. But we expect this number to be small. For other datasets, such as SfM-120k, GLD, YFCC15M, ImageNet, and LVD-142M, our FORB does not overlap with them. For the training data used by CLIP, since it is not publicly available, it is difficult to determine if our FORB benchmark shares duplicates with it.
> >
> > In practice, we believe such duplicates only have little impact on the performance evaluations. This data overlap issue was also discussed in the LAION-5B paper (see the 2nd paragraph of page 10):
> >
> > > Overall, we do not consider potential test set overlap to be a serious threat for the validity of results obtained with LAION-5B. OpenAI encountered the same question in the context of their pre-training dataset for CLIP and found only few examples of substantial performance difference due to data overlap on downstream target datasets.
> >
> > Therefore, we believe our conclusions drawn from Table 3 still hold. To make the descriptions in our paper more clear, in Table 3 we mark the results from CLIP, BLIP, BLIP2, and DiHT as “upperbound” performance, as their training data may overlap with FORB. We have also updated Table 2 to include more details.
> >
> > Finally, since FORB may share duplicates with existing training datasets (e.g., LAION-438M), which makes it not a real OOD benchmark, we further remove the duplicate images shared with LAION-5B, giving a deduplicate subset of FORB. We also evaluate the baselines on this subset and report the results in Tables 3-6 of the supplementary material. It can be observed that the results on the subset are very similar to those from the original FORB dataset. This further demonstrates that duplicate images tend to have little impact on the evaluations.
> >
> > For a new method developed in the future, if it is going to evaluate on our FORB benchmark, we suggest it firstly remove the potentially overlapped images from its training data with respect to FORB, and then train the model on the deduplicated training data.
> >
> > We have added the above discussions in Section A.4 of the supplementary material.
> >
> > **Reference:**
> > - Christoph Schuhmann, et al. LAION-5B: An open large-scale dataset for training next generation image-text models. 2022.
> >
> > **Q4: Line13: It is unclear what heterogeneity means. Authors should clarify.**
> >
> > **A:** We use “heterogeneity” here to indicate our FORB dataset involves different domains, which have different properties and difficulties.
> >
> > **Q5: The paper should make a table to compare other existing datasets for image retrieval.**
> >
> > **A:** Thank you for the suggestion. In our original manuscript, we briefly discussed other existing datasets in the Related Work section. We have added a new table for comparing other datasets; please see Table 6 of the main paper.

---

> > ### Comment · Reviewer_XRLd · 2023-08-29
> >
> > Thanks for the responses, which have addressed most of my concerns. My issue Q1 still exists. As different feature extractors are trained on different datasets, there are two sets of factors changed: (1) feature extractor (i.e., model architecture) along with training loss and others, (2) datasets. Therefore, the comparisons among features and methods are not fair. I understand that training all the feature extractors on each of the datasets is prohibitively expensive, and doing so might not be needed for the dataset track paper. Therefore, I suggest authors add this discussion in the limitation section.

---

> > > ### Author Response · Authors · 2023-08-29
> > > **Thank you for the suggestion**
> > >
> > > Thank you for the suggestion. We will add more discussions in the paper as suggested.

---

### Official Review · Reviewer_srxZ · 2023-07-21
**The paper lacks the details of data annotation**

**Rating:** 6
**Confidence:** 4
**Correctness:** not clear. See Opportunities For Impr…
**Clarity:** the paper is well written.

**Strengths:**

- the dataset can be used for OOD.

- a metric t-mAP is introduced to evaluate existing methods.

- The existing methods are well categorized and analyzed, and the comparison results are beneficial to the development of related methods.

**Additional Feedback:**

I will change my rating based on the responses.

**Documentation:**

data access is available.

**Limitations:**

Limitations are described in Section 5.

**Opportunities For Improvement:**

My main concern is data collection. I think it is important for benchmarking the existing methods.

- data annotation. The factors of image-level (easy, medium, hard) are not quantitative. Are the annotations labeled manually? auto? by workers or authors? how to be consistent?

- no details on data split.

**Relation To Prior Work:**

the differences are described in Section 2 Related work.

**Summary And Contributions:**

This paper introduces a dataset for benchmarking image search methods. Their contributions include:

- a dataset for image retrieval including 8 types of flat objects and three levels.

- an evaluation metric extended by mAP.

- comparison and analysis of different existing approaches.

---

> ### Author Response · Authors · 2023-08-21
> **Thanks for your constructive feedback**
>
> **Q1: Data annotation. The factors of image-level (easy, medium, hard) are not quantitative. Are the annotations labeled manually? auto? by workers or authors? how to be consistent?**
>
> **A:** It is true that the factors of difficulty level are not quantitative and assigning difficulty levels to images can be a subjective process. In practice, to reduce bias and make the difficulty labels as precise as possible, we asked different annotators (workers) to manually annotate the difficulty for the same image and used majority voting to determine the final difficulty level. From the evaluations results in Table 3, we can see that the retrieval accuracies are quite consistent with difficulty levels for all methods, _i.e._, the accuracies are high on easy queries, whereas they are low on hard queries.
>
> Also it is worth noting that the difficulty level annotations are provided to allow a more detailed analysis on the performance of image retrieval methods. Such a difficulty breakdown serves as an auxiliary tool and signals the retrieval quality from different perspectives. Therefore, although the factors of difficulties are not quantitative, such difficulty labels are still useful for performance investigation.
>
> **Q2: No details on data split.**
>
> **A:** We split our FORB benchmark data into three sets: query images, index images, and distractor images. We provide detailed statistics about these three sets in Table 1 of the main paper. In our released FORB dataset, we provide metadata to indicate which set a given image belongs to. Also note that our FORB benchmark does not provide training images. It serves as a testbed supplementary to existing benchmarks (see the first two paragraphs of Section 3 of the main paper).

---

> > ### Comment · Reviewer_srxZ · 2023-08-29
> > **Thanks for the reply.**
> >
> > Thanks for the reply.
> > Based on the responses, I maintain my rating.

---

### Official Review · Reviewer_348u · 2023-07-21
**FORB: A Flat Object Retrieval Benchmark for Universal Image Embedding**

**Rating:** 6
**Confidence:** 3
**Clarity:** The paper is well-written and organized.

**Strengths:**

This article focuses on an interesting problem with a clear motivation. It proposes a new evaluation metric and a novel dataset in this benchmark.

**Additional Feedback:**

No

**Correctness:**

The benchmark provided in this submission is well-constructed and thoughtfully designed.

**Documentation:**

Yes

**Ethics:**

The submission provides a comprehensive benchmark and does not involve the construction of a new dataset. Therefore, I think that there are no significant ethical concerns in this submission and further discussion or review is not required.

**Limitations:**

1. We cannot intuitively analyze the performance during training. In this benchmark, the author may consider adding some.
2.The article should introduce more experiments to prove the efficiency and performance of the benchmark under the same experimental settings.

**Opportunities For Improvement:**

This work should introduce more experiments to prove the stability and efficiency of this benchmark in the training process under the same architecture settings.

**Relation To Prior Work:**

NO

**Summary And Contributions:**

This work introduces a novel dataset for benchmarking visual search methods on flat images with diverse patterns for the generalization problem between domains. Moreover, it proposes a new evaluation metric motivated by matching score margin for offering a new perspective on image embedding quality.

---

> ### Author Response · Authors · 2023-08-21
> **Thanks for your constructive feedback**
>
> **Q: This work should introduce more experiments to prove the stability and efficiency of this benchmark in the training process under the same architecture settings.**
>
> **A:** Our work mainly focuses on introducing a new image retrieval benchmark and revealing the properties of different image retrieval strategies as well as their limitations. In particular, our evaluations of the performances for different retrieval methods do not involve a training process. More specifically, for both top-down and top-only methods, we use the fixed pretrained model weights from existing works and thus the inference is deterministic; for bottom-up methods, they are quite robust to the randomness in the construction of codebook. Therefore, to our knowledge the inference using each of the models under comparison is stable.

---

### Official Review · Reviewer_CwcQ · 2023-07-21
**Review for FORB: A Flat Object Retrieval Benchmark for Universal Image Embedding**

**Rating:** 9
**Confidence:** 5
**Clarity:** The paper is very well written and cl…

**Strengths:**

The proposed dataset is very interesting and helps to fill a significant gap in the realm of benchmarks that focus on object re-id in varying context. The effort to produce the dataset (through manual querying and curation) was significant and the per query annotation of difficulty as a function of occlusion, blur, truncation, color distortion, perspective distortion, texture complexity, and the area of the object in the query image is especially impressive and useful for image retrieval evaluation.

The paper is very clearly written and the evaluation and experiments are well documented and reasonable.

Figure 1 does an excellent job of communicating the types of queries that the FORB dataset encompasses.

The evaluation, while not completely exhaustive in terms of the embedding models and features evaluated, is more than sufficient to provide a baseline for the benchmark.

**Additional Feedback:**

I think this is a very interesting and well presented new benchmark that would be of significant interest to the broader image retrieval and object re-ID communities. I highly recommend its acceptance.

**Correctness:**

I have no concerns about correctness of the dataset construction or the experiments and evaluation.

**Documentation:**

I would like to see a clearer explanation of the criteria for assigning difficulty levels to images, and a definition of distractor images and how they were found added. Otherwise, I think there is sufficient documentation about data collection and organization and availability and maintenance, and sufficient information to support reproducability.

**Ethics:**

There is no discussion of ethical issues or use, but I am not concerned about that for this particular benchmark.

**Limitations:**

The idea of a benchmark that consists of no training data is interesting -- I worry some that this will limit adoption of the proposed approach by the broader metric learning community (which focuses more on task-specific training than the local feature matching community). I can see it being especially interesting for zero-shot evaluation and general embedding or local feature evaluation, however, so I'm not sure this is a huge limitation.

**Opportunities For Improvement:**

I think the exact description of a "hard" task is interesting -- while the example in Figure 1(h)(n) certainly is includes significant background noise, the card itself is an extremely clear match in all other regards. It would be helpful to see the specific criteria that was used when determining the difficulty of examples.

The paper does not currently give a clear definition for distractor images, and would benefit from that definition being added alongside example query-index-distractor pairs (perhaps in Figure 1, or in a new Figure).

**Relation To Prior Work:**

The authors do a good job of explaining how the proposed dataset and benchmark relates to and differs from existing work. I do research in this space and would be very excited to use this benchmark.

**Summary And Contributions:**

This paper introduces the Flat Object Retrieval Benchmark (FORB), which query images of eight different types of flat objects (animated trading cards, photorealistic trading cards, book covers (in different languages), paintings, currency, logos, packaged goods and movie posters).

The provided dataset includes query, index and distractor images. It notably does not contain any explicit training dataset, but rather is intended to be used for evaluation of existing or general purpose image retrieval models/embeddings.

---

> ### Author Response · Authors · 2023-08-21
> **Thanks for your constructive feedback**
>
> **Q1: It would be helpful to see the specific criteria that was used when determining the difficulty of examples.**
>
> **A:** The specific difficulty level of a query image is determined based on the following factors: (1) occlusion; (2) blur; (3) truncation; (4) color distortion; (5) perspective distortion; (6) texture complexity; (7) area of the object in the query image. For example, if the target object only occupies a small area in the image, we tag “hard" for the given query image due to the distraction of background; see Figure 1(h)(n). Similarly, if the target object does not bear severe perspective distortion or truncation, we tag “easy” for the query image; see Figure 1(b).
>
> In practice, since assigning difficulty levels to images can be a “subjective” process, to reduce bias and make the difficulty labels as precise as possible, we asked different annotators to manually annotate the difficulty for each image and used majority voting to determine the final difficulty level. It is also worth mentioning that from the evaluations results in Table 3, we can see that the retrieval accuracies are consistent with difficulty levels for all methods, i.e., the accuracies are high on easy queries, whereas they are low on hard queries.
>
> We have added more details on how to determine the difficulty level in the main paper (please see the first paragraph of Section 3.2).
>
> **Q2: The paper would benefit from giving a clear definition of distractor images.**
>
> **A:** Thank you for the suggestion. The distractor images are those that share similar semantics, contents, or textures with the index images. They can be from the same domains as the index images, or from other domains. Distractor images are primarily introduced to increase the retrieval difficulty, as they would bring perplexing features that deceive retrieval algorithms and reduce the accuracy of retrieval results. Ideally, a strong retrieval algorithm should be robust against distractor images.
>
> In our FORB dataset, distractor images bear similarities to the index images in various aspects. For example:
> 1. They have similar contents or textures. For instance,
>     - [index image](https://assets.pokemon.com/assets/cms2/img/cards/web/SWSHP/SWSHP_EN_SWSH045.png) VS [distractor image](https://assets.pokemon.com/assets/cms2/img/cards/web/SWSH3/SWSH3_EN_117.png).
>     - [index image](http://www.banknote.ws/COLLECTION/countries/AME/SUR/SUR0166o.jpg) VS [distractor image](http://www.banknote.ws/COLLECTION/countries/AME/SUR/SUR0163o.jpg).
> 2. They have different contents but in similar style:
>     - [index image](https://www.grandsudinsolite.fr/client/gfx/photos/produit/12-picasso-homme_17036.jpg) VS [distractor image](https://www.reproduction-gallery.com/catalogue/uploads/1463032273_large-image_pablopicassotheartistandhismodel1963lg.jpg).
> 3. Their contents bear the same semantics. This would pose challenges for top-only methods since their embeddings capture more about high-level semantics. For example, they both refer to the same movie:
>     - [index image](https://encrypted-tbn1.gstatic.com/images?q=tbn:ANd9GcTdW7z-a-2sI42LD5nqqaoIbHisnmWNaXec8pK_GQ6ymxrSAmqx) VS [distractor image](https://static1.srcdn.com/wordpress/wp-content/uploads/2020/12/Mad-Max-Fury-Road-Furiosa-poster.jpg?q=50&fit=crop&w=300&h=445&dpr=1.5).
>
>     Another example is that both distractor and index images refer to the same person:
>     - [index image](https://beckett-www.s3.amazonaws.com/news/news-content/uploads/2021/06/2005-06-SP-Game-Used-Chris-Paul-RC.jpg) VS [distractor image](https://beckett-www.s3.amazonaws.com/news/news-content/uploads/2021/06/2005-06-Upper-Deck-Reflections-Chris-Paul-RC.jpg).
> 4. Their contents share the same texts, which would pose challenges for text-sensitive methods such as CLIP.
>     - [index image](https://images-na.ssl-images-amazon.com/images/I/91p1kVwPrbL.jpg) VS [distractor image](http://prodimage.images-bn.com/pimages/9780778330905_p0_v7_s1200x630.jpg).
> 5. They have different contents but with similar shapes:
>     - [index image](https://logos-world.net/wp-content/uploads/2021/02/Peloton-Emblem.png) VS [distractor image](https://upload.wikimedia.org/wikipedia/commons/0/08/Pinterest-logo.png).
> 6. They are very similar products:
>     - [index image](https://1.bp.blogspot.com/-M5C3bzqbBmE/XXHH6oj89fI/AAAAAAAAo20/3ArHQXidBAg1qNBHONTxQJw6ulfNN8aTACLcBGAs/s640/mochi.jpg) VS [distractor image](https://cdn.yamibuy.net//item/c3c6f1d11062be6dccabbfde1406194a_757x757.webp).
>
> We have added the above examples in Figure 9 of the supplementary material. Also we have added the definition of distractor images in the last paragraph of Section 3.1 of the main paper.
>
> Finally, note that in our work we also utilize distractor images to compute the $t$-mAP metric.

---

> > ### Author Response · Authors · 2023-08-21
> > **Thanks for your constructive feedback**
> >
> > **Q3: FORB is especially interesting for zero-shot evaluation and general embedding or local feature evaluation. Not sure if the lack of training data will limit the adoption of FORB by the broader metric learning community.**
> >
> > **A:** While our FORB benchmark does not offer training data, it can still be used for evaluating the quality of learned metrics, i.e., as a zero-shot evaluation dataset. In particular, many methods in the metric learning community adopt image retrieval datasets for performance evaluation, such as CUB [48], Cars196 [20], and SOP [31]. This is because the learned metrics can be naturally employed to rank the database images based on their distances to the query image, and the retrieval accuracy reflects the quality of learned metrics. Therefore, we believe our FORB dataset will serve as a useful testbed for assessing the performance of learned metrics on out-of-distribution domains and thus can be adopted by the metric learning community.

---

> > > ### Comment · Reviewer_CwcQ · 2023-08-28
> > >
> > > These all sound like great updates. Thanks! I maintain my rating -- I think this is a really cool benchmark and am excited to hopefully see it published!

---

### Official Review · Reviewer_Utwz · 2023-07-21
**Good dataset for image retrieval.**

**Rating:** 7
**Confidence:** 3
**Correctness:** The claims are technically sound.
**Clarity:** The paper is well written overall.

**Strengths:**

- The authors introduce a new large-scale dataset for evaluating image retrieval methods. This FORB dataset has a unique characteristic because it contains 2D flat images while other datasets are mainly built using only 3D objects.

- The authors provide two new evaluation datasets to faitfully measure the performances of evaluated methods.

- The authors re-evaluate various retrieval methods with the proposed dataset and metrics, which is helpful for future researchers and practitioners.

**Additional Feedback:**

N/A

**Documentation:**

Yes.

**Limitations:**

The authors discuss the limitation of the proposed evaluation protocol well.

**Opportunities For Improvement:**

- It would be great if the authors provide their own baseline method, rather than just comparing existing methods.


**Relation To Prior Work:**

Yes.

**Summary And Contributions:**

This paper proposes a dataset for evaluating image retrieval methods. Since previous benchmark focuses only on 3D object, it is not effective to measuring the generalizability to different domain such as 2D images. To address this limitation, the authors introduce a new dataset, FORB, which contains 2D flat images. Also, they introduces two new evaluation metrics. With the dataset and metrics, the authors re-evaluate existing methods and based on it they discuss the current state of the research field.

---

> ### Author Response · Authors · 2023-08-21
> **Thanks for your constructive feedback**
>
> **Q: It would be great if the authors provide their own baseline method, rather than just comparing existing methods.**
>
> **A:** Thank you for the suggestion. In our paper, the comparisons among existing methods are provided to develop a better understanding of various retrieval strategies and thereby to gain insights into the development of new techniques. Based on our FORB dataset and experimental analysis, we are developing a new retrieval method that jointly leverages the mid- and high-level image features for more robust image retrieval. We will report this method in a follow-up manuscript. Besides, in the future we will continuously track new baseline methods and include them into our benchmark.

---

### Author Response · Authors · 2023-08-21
**Thank you for your constructive and helpful comments.**

We thank reviewers for constructive and helpful comments. Per their suggestions and questions, we have added more analysis and results in both the main paper and supplementary material. In particular, we have updated our main paper and supplementary material. Below we address the concerns of each reviewer one-by-one.

---

### Decision · Program_Chairs · 2023-09-22

**Decision:**

Accept (Poster)

**Comment:**

*Meta-review*

The reviewers agree that FORB is an addition to the field because of its scale and focus on 2D flat objects, and praise the benchmark's ability to measure generalization of a variety of models from a variety of datasets to this 2D domain. The proposed t-mAP metric is considered to be sensible.

Minor concerns were raised regarding the ability to draw conclusions from the baseline method results, given that the different models were trained on different datasets. But, given the overall positive sentiment, I recommend accepting this paper.

*Summary*

This work introduces a dataset and benchmark to measure the performance of image retrieval methods on "flat" images (e.g., logos, trading cards, paintings, etc.) The dataset consists of index images (flat, clean logos, trading cards, etc.), query images (noisier photos that are used to retrieve the index images), and distractor images (images similar to index images to increase retrieval difficulty). Query images are further annotated with their difficulty and source of difficulty (e.g., occlusion, perspective distortion, colour distortion).

The benchmark uses truncated mAP to measure the performance. A second metric, t-mAP, is introduced. This metric uses OOD queries (i.e., those with no corresponding index images) to calibrate the threshold needed to have a false positive rate of 0%, 10%, etc. The mAP scores are then calculated for the rankings returned by the method, after being thresholded.

The paper evaluates a series of off-the-shelf baseline models (using multi-scale features).